# Breast and cervical cancer screening among women at reproductive age in Cambodia: A secondary analysis of Cambodia Demographic and Health Survey 2022

**Samnang Um** [ID]*, **Heng Sopheab** [ID]

National Institute of Public Health, Phnom Penh, Cambodia

* umsamnang56@gmail.com

## Abstract

Breast and cervical cancers are the most prevalent diagnosed in women worldwide, significantly contributing to maternal morbidity and mortality. We examined socio-demographic and behavioral factors associated with breast and cervical cancer screening among Cambodian women aged 15–49 years old. We analyzed women's data from the 2022 Cambodia Demographic and Health Survey (CDHS). In total, 19,496 women were interviewed. Multiple logistic regression was performed using STATA V17 to examine factors associated with breast and cervical cancer screening. The proportion of breast and cervical cancer screenings was 10.6% and 15.3%, respectively. After being adjusted, factors independently associated with breast cancer screening included age group 20–29 years [AOR = 2.51; 95% CI: 1.55–4.06], 30–39 years [AOR = 4.34; 95% CI: 2.66–7.09], and 40–49 years [AOR = 4.66; 95% CI: 2.81–7.71], higher education [AOR = 1.92; 95% CI: 1.26–2.93], exposure media [AOR = 1.66; 95% CI: 1.32–2.10], and rich wealth quintile [AOR = 1.50; 95% CI: 1.25–1.80]. Similarly, the odds of having cervical cancer screening were age group 20–29 years [AOR = 2.88; 95% CI: 1.76–4.71], 30–39 years [AOR = 5.94; 95% CI: 3.58–9.83], and 40–49 years [AOR = 7.61; 95% CI: 4.55–12.73], higher education [AOR = 1.55; 95% CI: 1.55–2.73], exposure media [AOR = 1.62; 95% CI: 1.35,1.95], and rich wealth quintile [AOR = 2.14; 95% CI: 1.78–2.5f8]. In conclusion, this study shows the low screening proportion for both breast and cervical cancers, and it also highlights that socio-economic factors are significantly important in determining the health care seeking for these two main cancer screening services among women aged 15–49 years in Cambodia. Therefore, increase of exposure media with health education focusing on these cancer screenings should be made better accessible to women, particularly those of low socio-economic status.

## Introduction

Cancer is the most significant public health problem worldwide, with approximately 18.1 million cases (8.8 million in women) and 10 million cancer-related mortality reported in 2020 [1].

**Data Availability Statement:** Our study used the 2021-2022 Cambodia Demographic and Health Survey (CDHS) datasets. The DHS data are publicly available from the website at (URL:https://www.

dhsprogram.com/data/available-datasets.cfm). The Shapefiles for administrative boundaries in Cambodia are publicly accessible through the DHS website at (URL:https://spatialdata.dhsprogram.com/boundaries/#view=table&countryId=KH).

**Funding:** The author(s) received no specific funding for this work.

**Competing interests:** The authors have declared that no competing interests exist.

**Abbreviations:** ACS, American Cancer Society; AOR, Adjusted odds ratio; BMI, Body mass index; CDHS, Cambodia Demographic Health Survey; EA, Enumeration areas; HPV, Human papillomavirus; NCDs, Noncommunicable diseases; PPS, Probability proportional to size; VIA, Visual inspection with acetic acid; WRA, Women at reproductive age; WHO, World Health Organization.

Globally, breast and cervical cancers are the most prevalent diagnosed in women [2], especially since cervical cancer is a relatively more significant problem in low and middle-income countries (LMICs) than breast cancer [3]. According to estimates, cervical cancer accounted for more than 604,000 new cases and 342,000 deaths globally in 2020 [4], while breast cancer affected over 2.3 million women and resulted in 685,000 deaths; these most diagnosed cancers worldwide in women at any age after puberty, but with increasing rates when women get older [4, 5]. Evidence suggests that early diagnosis is pivotal in preventing and intervening in these cancers [6]. Early community-based screening and awareness programs have been in place in many developed countries, such as the United States, Finland, Sweden, France, and Japan [7–10]. Recently, national screening programs for cervical cancer have been widely provided in Asian countries, including China, Indonesia, Japan, Korea, and Thailand [11]. However, breast cancer screening as a national program has been slowly implemented due to the limited availability of screening guidelines across Asia, access barriers, budgeting constraints, stigma, fear of diagnosis, limited knowledge of breast self-examination, and cultural concerns [5, 12].

Evidence from LMICs such as Cambodia indicates that incomplete treatment and insufficient follow-up are the leading causes of the low proportion of breast and cervical cancer screening, with late detection resulting in high mortality. Other significant contributing factors have included social stigma, negligence, improper referral pathways, a lack of essential health infrastructure facilities, and ineffective treatment [13, 14]. In these settings, there are surprisingly few feasible and easily accessible screening programs despite the rising number of instances of breast and cervical cancers. With limited resources, many countries have adopted varying ages for breast and cervical cancer screenings. For instance, the minimum recommended age for breast cancer screening in Vietnam is 20 years, 30 years in India, 35 years in Sri Lanka, and 40 years each in China and Pakistan [15–18]. In the case of cervical cancer, China recommends 18 years as the minimum age for screening, while it is 20 years in Korea, 30 years each in India and Indonesia, and 35 years in Thailand [11]. Despite these guidelines, the screening prevalence is still low. For instance, cervical cancer screening ranges from 7.3% in Indonesia to 22.3% in India. Lack of screening knowledge, with socio-demographic, economic, cultural, and structural barriers are the main determinant for low screening in LMICs [19]. In Cambodia, breast and cervical cancers are remarkably rising in women in which the burden of breast cancer was 18.9% (1,877 cases), and cervical cancer was 11.4% (1,135 cases) reported in 2020 [1]. The age-standardized incidence rate (13.5 per 100,000 women) and the mortality rate (10.1 per 100,000 women) are much higher than regional and global estimates [1]. The recent data from CDHS 2021–22 have shown that only 11% of women aged 15–49 reported that healthcare had ever examined them to check for breast cancer and 15% reported that they had been tested for cervical cancer [20]. Breast and cervical cancers are increasing and become a public health problem [20]. Given these public health concerns, the breast and cervical cancers were included in the disease priority list in the National Strategy for the Prevention and Control of Non-Communicable Diseases (NCDs) 2007–2010 [21]. In 2008, the Cambodian Ministry of Health (MoH) issued a national guideline recommend for cervical cancer screening with visual inspection with acetic acid (VIA) for all sexually active women at every two years and immediate treatment with cryotherapy "e.g. screen and treat"). The new national action plan for cervical cancer prevention and control 2019–2023, has integrated HPV vaccination into the national program to support the prevention of cervical cancer in Cambodia [22]. Also, the new national action plan has addressed the screening methods for breast cancer including breast self-examination, and physical examination and mammography. However, it is expensive to test for Human papillomavirus (HPV), and it is less sensitive and less accurate than polymerase chain reaction (PCR)-based test used in more affluent settings [23]. Therefore, Cambodia has limited resources for implementing its strategic plan since

it still has many competing healthcare priorities, including infectious diseases, NCD, universal health coverage and quality of health [23]. In 2017–2018, there was an HPV vaccine campaign demonstration project to administer two doses of bivalent HPV vaccine to all 9-year-old girls residing in Svay Rieng and Siem Reap provinces; however, it is not yet widely available in Cambodia [24, 25].

According to existing literature, factors associated with cervical and breast screening among women include higher socioeconomic position (education, occupation and income) [26–29], ethnicity, age, marital status, media exposure [29], increased access to health care [27]; covered by health insurance [29, 30], urban residence [26, 29], positive lifestyle behaviors such as physical activity and fruit and vegetable consumption [30], cultural taboos, stigma, geographic conditions, and gender inequality [29]. In the context of limited resource countries such as Cambodia, in this study we use the social determinants of health theorical framework as the potential factors influence women awareness and utilization of health services such as breast and cervical cancer screening [31]. To our knowledge, very limited research has been observed in the area of breast and cervical cancer screening and its associated risk factors among Cambodian women. Thus, this paper aimed to assess the breast and cervical cancer screening services linked to women aged 15–49 and its determinants.

## Materials and methods

### Data sources

We used existing women's data from the latest 2022 CDHS, a nationally representative population-based household survey implemented by the National Institute of Statistics (NIS) in collaboration with MoH. Data was collected from September 15, 2021, to February 15, 2022 [20]. Two-stage stratified cluster sampling was used to collect the samples. First, selected 709 clusters or enumeration areas (EAs) stratified by urban-rural using probability proportional to cluster size. Second, selected 25–30 households in each EA using systematic sampling. In total, 19,496 women aged 15–49 were interviewed face-to-face, using the survey questionnaire, covering socio-demographic characteristics, alcohol drinking, tobacco use, household assets maternal health-related indicators and nutritional status, and data on breast and cervical cancer screening by healthcare providers [20]. The response rate was reported to be 98.2%. Overall, 254 women reported missing for breast and cervical cancer screening were excluded. Data restriction resulted in a final analysis with 19,251 weighted samples of women.

### Measurements

**Outcome variables.** The two separate outcome variables consisting of breast cancer and cervical cancer screening were defined based on the question 'Has a doctor or other healthcare provider examined your breasts to check for breast cancer, and 'Has a doctor or other healthcare worker ever tested you for cervical cancer?' Then the two outcomes were coded as binary variables with Yes = 1 to indicate cancer screening, and No = 0 to indicate otherwise, respectively.

**Independent variables.** Independent variables consisted of socio-demographic characteristics and behavioral factors: Women's age in years (15–19, 20–29, 30–39, and 40–49), marital status (never-married, currently married, and formerly married), education (no education, primary, secondary, and higher), occupation (not working, professional/formal, and non-professional/informal), and household wealth index reflected to the household assets and dwelling characteristics was categories as poorest, poorer, medium, richer, and richest. Provinces were grouped into four geographical regions (Plains, Tonle Sap, Coastal/sea, and Mountains) and place of residence (rural vs. urban) [20]. Health insurance coverage (yes vs. no), health facility

visits within the past year (yes vs. no), number of children ever born (no children, one-two children, three or more children). Current smoking was measured at the time of the survey (non-smoker vs. smoker, including daily smoker and occasional smoker). Alcohol consumption in the last 30 days (never drink, ever drink, and current drink which corresponds to one can or bottle of beer, one glass of wine, or one shot of spirits in the last 30 days, and contraceptive uses (hormonal methods and non-hormonal and traditional methods). Media exposure variables were "frequency of reading magazine or newspaper", "frequency of listening to radio", and "frequency of watching television", which were used to calculate the "media exposure score". Each variable was coded as 0 = not at all, 1 = less than once a week, and at least once a week; the computed "media exposure" index ranged from 0 to 3 and was categorized as (0 = no exposure media, 1 = one type of media, and 2 = two and more types of media). Body mass index (underweight ($< 18.5$ kg/m$^2$), normal weight (18.5–24.9 kg/m$^2$), overweight, and obesity ($\geq 25.0$ kg/m$^2$).

**Statistical analysis.** Statistical analyses were performed using STATA version 17 (Stata Crop 2019, College Station, TX). We accounted for CDHS sampling weight and complex survey design using the *survey package* in our descriptive and logistic regression analyses. Key socio-demographic characteristics and behavioral factors were described using weighted frequency. Continuous variables were calculated as mean (SD) such as women age. Bivariate analysis with chi-square tests was used to assess the association between the variables of interest (socio-demographic characteristics and behavioral factors) and breast and cervical cancer screening separately. Variables associated with breast and cervical cancer screening with at p-value $\leq 0.10$ were included in the multiple logistic regression analysis to determine the independent factors associated with the screening for breast and cervical cancer [32]. Multicollinearity between independent variables was checked including women's age, number of children ever born, education, wealth index, occupation, marital status, and place of residence. Variance inflation factor (VIF) scores were 1.53 indicated no collinearity concerns [33].

**Ethical consideration.** The study used women's data extracted from the most recent CDHS 2022 publicly available and granted through the DHS program (https://dhsprogram.com/data/available-datasets.cfm) with all removed personal identifiers. Written informed consent was obtained from the parent/guardian of each participant under 18 years of age before data collection. The Cambodia National Ethics Committee for Health Research approved the data collection tools and procedures for CHDS 2022 on 10 May 2021 (Ref # 83 NECHR) and ICF's Institutional Review Board (IRB) in Rockville, Maryland, USA.

# Results

## Characteristics of the study samples

Women's socio-demographic characteristics and behavioral factors are described in **Table 1**. The mean age was slightly close to 31 years old (SD = 9.5 years); the age group 30–49 years old accounted for 56.3%. More than 69.0% were currently married. Close to 43.0% of women worked in non-professional or informal jobs, and 27.2% were unemployed. Half of the women completed at least secondary education, while 11.5% had no formal education. Over one-fourth of women (26.9%) had three or more children. Close to one fourth of them (24.2%) reported using hormonal contraceptives. Over half (53.6%) of the women did not have exposure to media, and 28.7% were overweight and obese. Less than 2.0% of them reported cigarette smoking, and 16.20% currently drink alcohol. Of the total sample, 35.3% of women were from poor households. Over half (57.5%) of the women resided in rural areas.

**Table 1. Socio-demographic and behavioral characteristics of the weighted samples of women aged 15–49 years old in Cambodia, 2022 (N = 19,251).**

| Variables | | Freq. | Percent |
|---|---|---|---|
| **Mean age in years (±SD)** | | 30.8 (± 9.5) | |
| | 15–19 | 2,910 | 15.12 |
| | 20–29 | 5,506 | 28.60 |
| | 30–39 | 6,581 | 34.19 |
| | 40–49 | 4,253 | 22.09 |
| **Marital status** | | | |
| | Never married | 4,683 | 24.33 |
| | Currently married | 13,361 | 69.40 |
| | Formerly married | 1,206 | 6.26 |
| **Occupation** | | | |
| | Non-professional/Informal | 8,164 | 42.41 |
| | Professional/Formal | 5,852 | 30.40 |
| | Not working | 5,234 | 27.19 |
| **Education** | | | |
| | No education | 2,226 | 11.56 |
| | Primary | 7,458 | 38.74 |
| | Secondary | 8,178 | 42.48 |
| | Higher | 1,389 | 7.22 |
| **Number of children born** | | | |
| | No birth | 5,693 | 29.57 |
| | 1–2 children | 8,385 | 43.56 |
| | ≥ 3 children | 5,173 | 26.87 |
| **Contraceptive use** | | | |
| | Not use | 10,909 | 56.67 |
| | Hormonal | 4,660 | 24.21 |
| | Non-hormonal | 3,681 | 19.12 |
| **Media exposure (Newspaper, radio, TV)** | | | |
| | None | 10,319 | 53.60 |
| | One | 6,103 | 31.70 |
| | ≥ Two | 2,829 | 14.70 |
| **BMI (n = 9,708)** | | | |
| | Underweight | 1,001 | 10.31 |
| | Normal | 5,918 | 60.96 |
| | Overweight/Obese | 2,789 | 28.73 |
| **Alcohol drinking in the past 30 days** | | | |
| | Never drink | 12,548 | 65.18 |
| | Ever drink | 3,583 | 18.61 |
| | Current drink | 3,119 | 16.20 |
| **Smokes cigarettes** | | | |
| | Non-smoker | 18,967 | 98.52 |
| | Smoker | 284 | 1.48 |
| **Health insurance coverage** | | | |
| | No | 15,063 | 78.25 |
| | Yes | 4,188 | 21.75 |
| **Wealth index** | | | |
| | Poor | 6,787 | 35.26 |

*(Continued)*

**Table 1.** (Continued)

| Variables | | Freq. | Percent |
|---|---|---:|---:|
| | Middle | 3,788 | 19.68 |
| | Rich | 8,676 | 45.07 |
| **Place of residence** | | | |
| | Urban | 8,190 | 42.54 |
| | Rural | 11,060 | 57.45 |
| **Geographical region\*** | | | |
| | Plain | 9,723 | 50.51 |
| | Tonle Sap | 5,744 | 29.84 |
| | Plateau/Mountain | 2,590 | 13.45 |
| | Coastal | 1,194 | 6.20 |

**Notes:** Survey weights were applied to obtain weighted percentages. \***Plains:** Phnom Penh, Kampong Cham, Tbong Khmum, Kandal, Prey Veng, Svay Rieng, and Takeo; **Tonle Sap:** Banteay Meanchey, Kampong Chhnang, Kampong Thom, Pursat, Siem Reap, Battambang, Pailin, and Otdar Meanchey; **Coastal/sea:** Kampot, Kep, Preah Sihanouk, and Koh Kong; **Mountains:** Kampong Speu, Kratie, Preah Vihear, Stung Treng, Mondul Kiri, and Ratanak Kiri.

## Prevalence of breast and cervical screening among women and provincial distribution

Overall, 10.6% (95% CI: 9.7%-11.6%) of women had breast cancer screening by health providers, whereas cervical cancer screening was 15.3% (95% CI: 14.2%-16.6%). **Fig 1** indicates the province specific prevalence of breast cancer screening **(Map A)** and cervical cancer screening **(Map B)**. The prevalence of breast cancer screening was the highest among women in Siem Reap (17.6%), followed by Phnom Penh (17.2%), Battambang (17.1%), Kampong Thom (15.3%), and the lowest in Kep (2.6%). The prevalence of cervical cancer screening was the highest among women in Phnom Penh (28.5%), followed by Kampong Thom (21.7%), Battambang (20.1%), and the lowest in Kampong Speu (3.1%).

## Factors associated with breast cancer screening in bivariate analysis

**Table 2** shows the association between socio-demographic characteristics and behavioral factors and breast cancer screening. Women had a higher proportion of breast cancer screening among those aged 40–49 years (14.4%) than others (p-value <0.001). The women with professional jobs (15.2%) had higher education (22.4%) than other job categories (p-value <0.001). Also, women reported at least three children had a significantly higher experience with breast cancer screening (14.0%) than those with fewer children (p-value < 0.001); likewise, breast cancer screening was higher among women using non-hormonal contraceptives (18.7%) than those with hormonal use (p-value <0.001). Furthermore, following factors were found significantly associated with breast cancer screening: exposure to at least two types of media (16.0%) overweight and obese (14.2%) (p-value <0.001), report of never drinking alcohol (11.3%) (p-value = 0.003), non-smoker (11.3%) (p-value = 0.049), rich wealth quintile households (14.7%) (p-value <0.001), higher in urban areas than rural ones (13.1% vs 8.7%, p-value <0.001).

## Factors associated with cervical cancer screening in bivariate analysis

The study found a strong association between socio-demographic characteristics and behavioral factors and cervical cancer screening, as indicated in **Table 2**. Among women aged 40–49

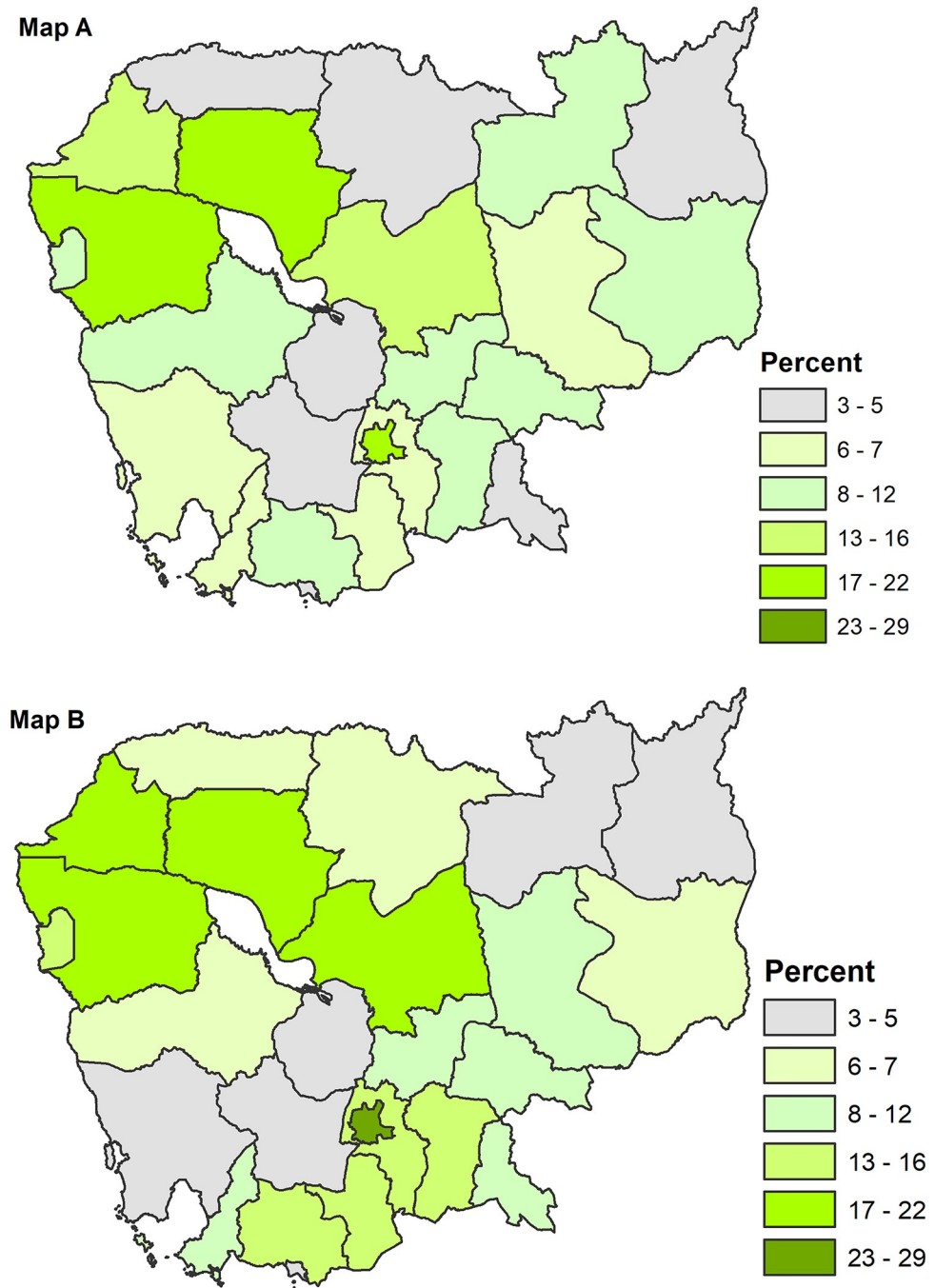

**Fig 1.** Prevalence of breast cancer screening (**Map A**) and cervical cancer screening (**Map B**) among women by province. Shapefiles for administrative boundaries in Cambodia are publicly accessible through the DHS website at (UR:https://spatialdata.dhsprogram.com/boundaries/#view=table&countryId=KH).

years, there was a significantly greater proportion of breast cancer screening (23.2%) compared to other groups (p-value <0.001). This pattern was observed among formerly married women (20.2%) (p-value <0.001), those with professional occupations (23.7%), and those with higher levels of education (23.5%) (p-value <0.001). Also, women reported at least three children had a significantly higher experience with breast cancer screening (21.0%) than women with

**Table 2. Factors associated with breast and cervical cancer screening among women aged 15–49 years (N = 19,251).**

| Variables | (%) Breast cancer screening (n = 2,039) | $X^2$ | P-value | (%) Cervical cancer screening (n = 2,952) | $X^2$ | P-value |
|---|---|---|---|---|---|---|
| **Women's age in years** | | | | | | |
| 15–19 | 1.4 | 492.6 | <0.001 | 1.0 | 989.4 | <0.001 |
| 20–29 | 7.7 | | | 9.6 | | |
| 30–39 | 14.7 | | | 21.4 | | |
| 40–49 | 14.4 | | | 23.2 | | |
| **Marital status** | | | | | | |
| Never-married | 2.3 | 451.1 | <0.001 | 1.8 | 876.1 | <0.001 |
| Currently married | 13.2 | | | 19.7 | | |
| Formerly married | 14.0 | | | 20.2 | | |
| **Occupation** | | | | | | |
| Professional/Formal | 15.2 | 192.4 | <0.001 | 23.7 | 465.3 | <0.001 |
| Non-professional/Informal | 8.8 | | | 12.4 | | |
| Not working | 8.1 | | | 10.5 | | |
| **Education** | | | | | | |
| No education | 8.8 | 45.3 | <0.001 | 12.9 | 90.9 | <0.001 |
| Primary | 10.8 | | | 15.8 | | |
| Secondary | 10.0 | | | 14.2 | | |
| Higher | 15.5 | | | 23.5 | | |
| **Number of children born** | | | | | | |
| No birth | 3.4 | 437.0 | <0.001 | 4.0 | 811.9 | <0.001 |
| 1–2 children | 13.4 | | | 19.5 | | |
| ≥ 3 children | 14.0 | | | 21.0 | | |
| **Contraceptive use** | | | | | | |
| Not use | 7.9 | 305.7 | <0.001 | 11.0 | 715.8 | <0.001 |
| Hormonal | 11.1 | | | 14.6 | | |
| Non-hormonal | 18.1 | | | 29.3 | | |
| **Media exposure (Newspaper, radio, TV)** | | | | | | |
| None | 8.9 | 106.8 | <0.001 | 13.0 | 152.0 | <0.001 |
| One | 11.3 | | | 16.0 | | |
| Two or more | 15.5 | | | 22.3 | | |
| **BMI (n = 9,708)** | | | | | | |
| Underweight | 5.3 | 62.7 | <0.001 | 6.3 | 148.5 | <0.001 |
| Normal | 10.5 | | | 14.3 | | |
| Overweight/Obese | 14.2 | | | 21.6 | | |
| **Alcohol drinking in the past 30 days** | | | | | | |
| Never drink | 11.3 | 30.6 | 0.003 | 14.3 | 46.0 | <0.001 |
| Ever drink | 8.0 | | | 15.7 | | |
| Current drink | 10.8 | | | 19.2 | | |
| **Smokes cigarettes** | | | | | | |
| Non-smoker | 10.6 | 4.4 | 0.049 | 15.5 | 17.8 | <0.001 |
| Smoker | 6.8 | | | 6.4 | | |
| **Health insurance coverage** | | | | | | |
| No | 10.5 | 1.1 | 0.472 | 15.0 | 6.6 | 0.112 |
| Yes | 11.0 | | | 16.6 | | |
| **Wealth index** | | | | | | |
| Poor | 7.4 | 203.4 | <0.001 | 8.7 | 602.7 | <0.001 |

*(Continued)*

**Table 2.** (Continued)

| Variables | (%) Breast cancer screening (n = 2,039) | X² | P-value | (%) Cervical cancer screening (n = 2,952) | X² | P-value |
|---|---|---|---|---|---|---|
| Middle | 8.3 | | | 11.2 | | |
| Rich | 14.1 | | | 22.3 | | |
| **Place of residence** | | | | | | |
| Urban | 13.1 | 94.5 | <0.001 | 20.1 | 253.5 | <0.001 |
| Rural | 8.7 | | | 11.8 | | |
| **Geographical of region*** | | | | | | |
| Tonle Sap | 13.6 | 158.0 | <0.001 | 16.6 | 282.7 | <0.001 |
| Plain | 10.7 | | | 17.8 | | |
| Coastal | 8.7 | | | 12.0 | | |
| Plateau/Mountain | 4.6 | | | 4.8 | | |

**Notes:** Survey weights are applied to obtain weighted percentages. *Plains: Phnom Penh, Kampong Cham, Tbong Khmum, Kandal, Prey Veng, Svay Rieng, and Takeo; **Tonle Sap:** Banteay Meanchey, Kampong Chhnang, Kampong Thom, Pursat, Siem Reap, Battambang, Pailin, and Otdar Meanchey; **Coastal/sea:** Kampot, Kep, Preah Sihanouk, and Koh Kong; **Mountains:** Kampong Speu, Kratie, Preah Vihear, Stung Treng, Mondul Kiri, and Ratanak Kiri.

children less than 3 (p-value < 0.001). Similarly, the prevalence of breast cancer screening was higher among women who used non-hormonal contraceptives (29.3%) than those who used hormonal contraceptives (p-value < 0.001). Moreover, other factors that exhibited a significant association with cervical cancer screening included having exposure to at least two forms of media (22.3%), being overweight or obese (21.6%) (p-value <0.001), reporting current alcohol consumption (19.2%) (p-value 0.001), being a non-smoker (15.5%) (p-value 0.001), rich wealth quintile (22.3%) (p-value <0.001), and residing in urban areas as opposed to rural areas (20.1% vs 11.8%, p-value <0.001).

## Factors associated with breast cancer screening in multiple logistic regression

As shown in **Table 3**, several factors were independently associated with increased odds of having breast cancer screening among women. These factors included age group 20–29 years [AOR = 2.51; 95% CI: 1.55–4.06], 30–39 years [AOR = 4.34; 95% CI: 2.66–7,09], and 40–49 years [AOR = 4.66; 95% CI: 2.81–7.71], currently married [AOR = 2.67; 95% CI: 1.79–3.97], formerly married [AOR = 3.16; 95% CI: 1.93–5.17], completed secondary education [AOR = 1.44; 95% CI: 1.12–1.85], higher education [AOR = 1.92; 95% CI: 1.26–2.93], using non-hormonal contraceptive [AOR = 1.34; 95% CI: 1.08–1.65], exposure media at least two types [AOR = 1.66; 95% CI: 1.32–2.10], and rich wealth quintile [AOR = 1.50; 95% CI: 1.25–1.80]. On the contrary, women reporting ever alcohol consumption [AOR = 0.55; 95% CI: 0.45–0.66], and those reporting current alcohol drinking were associated with lower odds of having breast cancer screening [AOR = 0.75; 95% CI: 0.63–0.89].

## Factors associated with cervical cancer screening in multiple logistic regression

Similarly, following factors were independently associated with increased odds of having cervical cancer screening among women: age group 20–29 years [AOR = 2.88; 95% CI: 1.76–4.71], 30–39 years [AOR = 5.94; 95% CI: 3.58–9.83], and 40–49 years [AOR = 7.61; 95% CI: 4.55–12.73], currently married [AOR = 5.38; 95% CI: 3.52–8.21], formerly married [AOR = 5.69; 95% CI: 3.70–8.74], working in the professional and/or formal jobs [AOR = 1.38; 1.14–1.66],

**Table 3. Factors independently associated with breast and cervical cancer screening among women aged 15–49 years: Multivariate logistic regression.**

| Variables | | Breast cancer screening (N = 19,251) | | Cervical cancer screening (N = 19,251) | |
|---|---|---|---|---|---|
| | | AOR | 95% CI | AOR | 95% CI |
| **Women's age in years** | | | | | |
| | 15–19 | Ref. | Ref. | Ref. | Ref. |
| | 20–29 | 2.51*** | [1.55–4.06] | 2.88*** | [1.76–4.71] |
| | 30–39 | 4.34*** | [2.66–7.09] | 5.94*** | [3.58–9.83] |
| | 40–49 | 4.66*** | [2.81,7.71] | 7.61*** | [4.55–12.73] |
| **Marital status** | | | | | |
| | Never-married | Ref. | Ref. | Ref. | Ref. |
| | Currently married | 2.67*** | [1.79–3.97] | 5.38*** | [3.52–8.21] |
| | Formerly married | 3.16*** | [1.93–5.17] | 5.69*** | [3.70–8.74] |
| **Occupation** | | | | | |
| | Not working | Ref. | Ref. | Ref. | Ref. |
| | Professional/Formal | 1.20 | [0.99–1.47] | 1.38*** | [1.14–1.66] |
| | Non-professional/Informal | 0.89 | [0.73–1.09] | 0.94 | [0.79–1.12] |
| **Education** | | | | | |
| | No education | Ref. | Ref. | Ref. | Ref. |
| | Primary | 1.19 | [0.94–1.52] | 1.14 | [0.94–1.38] |
| | Secondary | 1.44** | [1.12–1.85] | 1.34** | [1.09–1.65] |
| | Higher | 1.92** | [1.26,2.93] | 2.06*** | [1.55–2.73] |
| **Number of children born** | | | | | |
| | No birth | Ref. | Ref. | Ref. | Ref. |
| | 1–2 children | 1.36 | [0.92–2.02] | 1.31 | [0.99–1.75] |
| | ≥ 3 children | 1.38 | [0.94–2.03] | 1.31 | [0.94–1.82] |
| **Contraceptive use** | | | | | |
| | Not use | Ref. | Ref. | Ref. | Ref. |
| | Hormonal | 0.97 | [0.80–1.19] | 0.83* | [0.70–0.98] |
| | Non-hormonal | 1.34** | [1.08–1.65] | 1.46*** | [1.23–1.73] |
| **Media exposure (Newspaper, radio, TV)** | | | | | |
| | None | Ref. | Ref. | Ref. | Ref. |
| | One | 1.20* | [1.04–1.39] | 1.09 | [0.96–1.24] |
| | ≥ Two | 1.66*** | [1.32–2.10] | 1.62*** | [1.35–1.95] |
| **Alcohol drinking in the past 30 days** | | | | | |
| | Never drink | Ref. | Ref. | Ref. | Ref. |
| | Ever drink | 0.55*** | [0.45–0.66] | 0.90 | [0.75–1.08] |
| | Current drink | 0.75** | [0.63–0.89] | 1.12 | [0.95–1.31] |
| **Smokes cigarettes** | | | | | |
| | Non-smoker | Ref. | Ref. | Ref. | Ref. |
| | Smoker | 0.72 | [0.43–1.19] | 0.43** | [0.26–0.72] |
| **Health insurance coverage** | | | | | |
| | No | Ref. | Ref. | Ref. | Ref. |
| | Yes | 0.96 | [0.82–1.13] | 1.00 | [0.86–1.17] |
| **Household wealth index** | | | | | |
| | Poor | Ref. | Ref. | Ref. | Ref. |
| | Middle | 1.00 | [0.82–1.23] | 1.14 | [0.95–1.36] |
| | Rich | 1.50*** | [1.25–1.80] | 2.14*** | [1.78–2.58] |
| **Place of residence** | | | | | |

*(Continued)*

**Table 3.** (Continued)

| Variables | | Breast cancer screening (N = 19,251) | | Cervical cancer screening (N = 19,251) | |
|---|---|---|---|---|---|
| | | AOR | 95% CI | AOR | 95% CI |
| | Rural | Ref. | Ref. | Ref. | Ref. |
| | Urban | 0.87 | [0.73–1.03] | 0.85* | [0.73–1.00] |

\* $p < 0.05$

\*\* $p < 0.01$

\*\*\* $p < 0.001$

completed secondary education [AOR = 1.34; 95% CI: 1.09–1.65], had higher education [AOR = 1.55; 95% CI: 1.55–2.73], using non-hormonal contraceptive [AOR = 1.46; 95% CI: 1.29–1.66], exposure media at least two types [AOR = 1.62; 95% CI: 1.35–1.95], and rich wealth quintile [AOR = 2.14; 95% CI: 1.78–2.58]. On the contrary, women reported smoking cigarettes was associated with lower odds of having cervical cancer screening [AOR = 0.43; 95% CI: 0.2–0.72] (**Table 3**).

## Discussion

The study highlights the relatively low proportion of breast cancer screening (10.6%), and cervical cancer screening (15.3%) respectively, compared to some Asian countries such as Thailand (46%-59%) and China (17%-26%) but comparable to Indonesia and India [11]. The study was consistent with a previous study that indicated that most women had performed more cervical cancer screening than breast cancer screening especially in low and middle-income countries [34]. The main reason may be due to the awareness and concern about cervical cancer among women rather than the breast cancer despite the prevalence of breast cancer being relatively higher than cervical cancer [35]. Also, it may be explained by the available national cancer screening guidelines focusing on cervical cancer screening [21, 36]. Additionally, awareness about breast self-examination (BSE) in Cambodia was limited. Approximately 60% of Cambodian women were unaware of BSE [37]. Moreover, cancer screening uptake remained low compared to screening rates of above 50% in developed countries [35]. Therefore, strengthening cancer screening in Cambodia requires ongoing efforts.

Overall, socio-economic factors were associated with breast and cervical cancer screening [38, 39]. For example, in our study, the odds of breast cancer screening increased with ages and the highest in age 40–49 years. This corroborates previous studies showing that women aged 40 years and above are at a greater risk of developing breast cancer [40, 41]. Consequently, women aged 35–39 should undergo routine mammography to enhance the early detection of breast cancer at least at the interval of 1–3 years and annually for women aged 40 years and older [41]. Similarly, the odds of being screened for cervical cancer increased with the women's age too, especially among those aged 40–49 years, which is commonly consistent with other studies after years of women's being sexually active [34, 42, 43]. The American Cancer Society (ACS) recommends that women aged 25 should undergo cervical cancer screening and be tested for HPV every five years through age 65 years [44]. Women with higher education levels were more likely to increase the odds of breast cancer and cervical screening than those without formal education, being consistent with previous studies [38, 39]. Moreover, contextual factors such as less educated women living in remote areas may contribute to the less screening [45, 46]. Similarly, women from rich families and professional or formal employment were more likely to have received breast or cervical cancer screening. These

findings are consistent with the results of other international studies [39, 47]. A possible reason could be that women with strong financial backing are more likely to afford preventive care services, including breast cancer and cervical cancer screening, which may not be the case for women of the lowest economic status [48]. Another finding worth commenting on is the higher odds of breast cancer and cervical cancer screening among women exposed to the media (e.g., Newspaper, radio, TV) at least once a week compared to those not exposed at all. Therefore, intensive publicity and educational programs on various mass media platforms on the health benefits of periodic breast examinations for early detection and treatment had the potential to improve breast cancer screening behavior among women by building on their awareness levels.

## Limitation

This study has a few limitations. First, the data were cross-sectional, preventing any causal inference between the explanatory and outcome measures. Secondly, since we used existing data, some variables must be added to the dataset—for instance, knowledge and attitude regarding cancer preventive services and frequency of screening. Thirdly, the examination of breast and cervical cancer data was collected as self-reported by the women, which may make our analysis prone to information bias as underestimated. Despite these limitations, this study is the first to assess the socio-economic and behavioral factors associated with breast and cervical cancer screening in a nationally-representative sample of women of reproductive age 15–49 years in Cambodia. Furthermore, the study provides important insights regarding the prevalence and socio-demographic factors of breast cancer and cervical screening services with the nationally representative sample. The findings from this study could be valuable for researchers, clinicians and health policymakers to engage more in developing effective strategies for promoting breast and cervical cancer screening services, awareness among women from different socio-demographic groups since it is has played a pivotal to improving early detection and screening practices. Health policymakers should prioritize health promotion interventions target women living in rural areas, women without formal education, especially in the poor wealth quintile.

## Conclusion

This study shows the relative lower screening proportion for both breast and cervical cancers among Cambodian women reproductive age; and it also highlights that socio-demographic factors are significantly important in determining the of breast and cervical screening services among women aged 15–49 years. The findings emphasize the need for targeted awareness campaigns, continued efforts in health promotion and education, and comprehensive interventions to address these socio-demographic disparities breast and cervical cancer screenings practices. Further studies are therefore necessary to generate a better picture of the situation, including the quality of the health service and factors associated with the accessibility of the available services.

## Supporting information

**S1 Data.**
(DTA)

## Acknowledgments

The authors would like to thank DHS-ICF, who approved the data use for this paper.

## Author Contributions

**Conceptualization:** Samnang Um, Heng Sopheab.

**Data curation:** Samnang Um.

**Formal analysis:** Samnang Um, Heng Sopheab.

**Investigation:** Samnang Um, Heng Sopheab.

**Methodology:** Samnang Um, Heng Sopheab.

**Project administration:** Samnang Um, Heng Sopheab.

**Software:** Samnang Um.

**Supervision:** Heng Sopheab.

**Validation:** Samnang Um, Heng Sopheab.

**Visualization:** Samnang Um, Heng Sopheab.

**Writing – original draft:** Samnang Um, Heng Sopheab.

**Writing – review & editing:** Samnang Um, Heng Sopheab.

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
