## [Decision Letter · Decision Letter 0]

21 Feb 2024

PONE-D-23-39984Breast and cervical cancer screening among women at reproductive age in Cambodia: Data analysis of Cambodia Demographic and Health Survey 2022PLOS ONE

Dear Dr. Um,

Thank you for submitting your manuscript to PLOS ONE. After careful consideration, we feel that it has merit but does not fully meet PLOS ONE’s publication criteria as it currently stands. Therefore, we invite you to submit a revised version of the manuscript that addresses the points raised during the review process.

We look forward to receiving your revised manuscript.

Kind regards,

Pijush Kanti Khan, Ph.D.

Academic Editor

PLOS ONE

Journal Requirements:

Reviewers' comments:

Reviewer's Responses to Questions

**Comments to the Author**

1. Is the manuscript technically sound, and do the data support the conclusions?

Reviewer #1: Partly

Reviewer #2: Yes

2. Has the statistical analysis been performed appropriately and rigorously? 

Reviewer #1: No

Reviewer #2: Yes

3. Have the authors made all data underlying the findings in their manuscript fully available?

Reviewer #1: No

Reviewer #2: Yes

4. Is the manuscript presented in an intelligible fashion and written in standard English?

Reviewer #1: No

Reviewer #2: Yes

5. Review Comments to the Author

Reviewer #1: Data 29/01/2024

To: PLOS ONE editorial team and Authors

Review comments

Dear editorial team

Thank you for inviting to review this articles. Dear authors you welcome with best and interesting research topic. The maternal health is one of the sustainable development goal which needs attention of the world program designer and policy makers. After I said this the comments of the manuscript put section by section below.

Review title: Breast and cervical cancer screening among women at reproductive age in Cambodia: Data analysis of Cambodia Demographic and Health Survey 2022

Abstract: The abstract could be outline as background, method, result and conclusion formats

Q1: Which one is one-third of all cancers worldwide? Is breast cancer or cervical cancer

Q2: Is demographic and health survey data incorporate behavioral factors that influence screening services? If so Which behavioural factors were included?

Q3: From which baseline the proportion of screening is low?

Q4: is media exposure or access to media data were collected by DHS survey?

Introduction: It good work

Q1: the national breast cancer and cervical cancer screening strategy, the current Cambodian policy about breast cancer and cervical cancer screening?

Q2: Previous literatures about factors of breast cancer and cervical cancer screening?

Q3: What is your theoretical framework of this study and briefly put your theoretical framework

Methods

Q1: 254 reproductive age women were missed, how did you manage this missing data?

Q2: Has a doctor or other healthcare worker ever tested you for cervical cancer is your outcome question, if the women tested twice, three times, which one did you take?

Q3: The independent variables could be listed for each outcome variables since the two outcome variables are quite different.

Q4: In DHS data there is clustering is so how did you manage this clustering in logistic regression analysis (why you conduct multilevel analysis)

Q5: What is the specific value of variance inflation factor (VIF)?

Q6: In one area it says logistic regression but in one area it says Ordinary Least Squares

regression model, which one is correct?

Q7: how did you check model fitness?

Q8: Is 19242 the sample size of both breast cancer and cervical cancer, if so how did you merge since the two outcomes were measured differently and it could have different missed value, different response and such like…please put the sample size of each outcome variables

Q9: Operational definition of some independent variables could be necessary like wealth index, media exposure…..

Q10: Is method of screening and frequency of screening not your concern?

Q11: is media access and media exposure the same in your study?

Q12: What about the women already positive for breast cancer, and cervical caner and women age above 49 years?

Results

Q1: The mean age were calculated but reported with standard deviation why?

Q2: In your method part, you wrote 19242 women were included for final analysis while in result section 19251 women included, why this discrepancy occur(19242 VS 19251)?

Q3: How smoking was measured (smoking in last ten years, is smoking last 20 years or smoking on data collection) which one?

Q4: In alcohol drinking you classify never drink, ever drink and currently drink if so currently drink is included in ever drink individuals, how did you manage it?

Q5: You calculated Odds ratio but you say risk factor, how?

Q6: There is new variable in your result section, how?

Discussion

Q1: You did not measure awareness about breast self-examination

Q2: Please write the implication of your finding?

Conclusion

Q1: write result specific conclusion and recommendation

Write the authors contribution of the study

Reviewer #2: The manuscript is very interesting and contains significant information on breast and cervical cancer screening across the reproductive age group and it is worthy for publication.

However, there are few minor suggestions which I would like to suggest.

1. In tittle, I would suggest that replace the "Data analysis" as "A secondary analysis/ A statistical analysis".

2. In Table-2, It is well understood with only Yes (%) out of total sample of breast and cervical cancer screening. Here you can add the chi-square value.

6. PLOS authors have the option to publish the peer review history of their article (what does this mean?). If published, this will include your full peer review and any attached files.

Reviewer #1: No

Reviewer #2: No

---

## [Author Response · Author response to Decision Letter 0]

26 Mar 2024

Authors’ point-by-point responses to the editor and reviewers

Manuscript ID: PONE-D-23-39984 

Manuscript Titled: Breast and cervical cancer screening among women at reproductive age in Cambodia: Data analysis of Cambodia Demographic and Health Survey 2022.

Dear editor and reviewers,

We would like to thank all the reviewers and editor for providing their important time and effort in reviewing our manuscript. We have addressed all comments where relevant, as listed below:

Reviewer #1:

Abstract: The abstract could be outline as background, method, result and conclusion formats

Author's response: The abstract was written to comply with the requirements for PLOS ONE format.

Q1: Which one is one-third of all cancers worldwide? Is breast cancer or cervical cancer

Authors’ response: We revised it and highlighted in the revised manuscript accordingly.

Q2: Is demographic and health survey data incorporate behavioral factors that influence screening services? If so Which behavioural factors were included?

Author’s response: Behavioral factors that influence screening of breast self-examination and cervical screening services were not included in the CDHS 2022 questionnaire.

Q3: From which baseline the proportion of screening is low?

Author's response: We compared with DHS in other countries, such as Nepal Demographic and Health Survey in 2022 and Kenya Demographic and Health Surveys in 2022.

Q4: is media exposure or access to media data were collected by DHS survey?

Authors’ response: For clarity, the CDHS collected women's exposure to some types of media, such as “Frequency of reading newspapers or magazines,” “Frequency of listening to radio,” and “Frequency of watching television.” We have revised and highlighted (Lines 162-167, Page 6).

Introduction:

Q1: the national breast cancer and cervical cancer screening strategy, the current Cambodian policy about breast cancer and cervical cancer screening?

Author’s response: We have updated and highlighted the revised manuscript by adding a National Action Plan for Cervical Cancer Prevention and Control 2019-2023. It has integrated HPV vaccination into the national immunization program to support cervical cancer prevention in Cambodia. Also, the new national action plan has addressed the screening methods for breast cancer included breast self-examination, physical examination and mammography. .

Q2: Previous literatures about factors of breast cancer and cervical cancer screening?

Authors’ response: We have revised and highlighted in the text accordingly (Line 91-96)

Q3: What is your theoretical framework of this study and briefly put your theoretical framework

Authors’ response: It has been based on the social determinants of health. We have revised and highlighted in the text Line 97-99. 

Methods

Q1: 254 reproductive age women were missed; how did you manage this missing data?

Authors’ response: The 254 women who reported “don’t known” were excluded from our analysis.

Q2: Has a doctor or other healthcare worker ever tested you for cervical cancer is your outcome question, if the women tested twice, three times, which one did you take?

Author’s response: in DHS, the women were ever asked for breast and cervical cancer screening. We have revised and highlighted in the revised manuscript for clarity for readers. 

Q3: The independent variables could be listed for each outcome variables since the two outcome variables are quite different.

Authors’ response: We do not agree with the reviewer since the same independent variables were used to assess association with the two outcomes. 

Q4: In DHS data there is clustering is so how did you manage this clustering in logistic regression analysis (why you conduct multilevel analysis).

Author’s response: We do not conduct multilevel analysis. We just use multiple logistic regression analysis, accounting for the complex survey design, as many published papers do such as: Chhea, C., et al. (2018) in PLOS ONE, Um, S., et al. (2023) in PLOS ONE, and Lamichhane, B., et al. (2024) in PLOS GLOBAL PUBLIC HEALTH.

Q5: What is the specific value of variance inflation factor (VIF)?

Author’s response: The use of VIF is to evaluate collinearity among independent variables. We have revised and highlighted the text accordingly in the revised manuscript (Line 167-170). 

Q6: In one area it says logistic regression but in one area it says Ordinary Least Squares regression model, which one is correct?

Author’s response: The main analysis is the multivariate logistic regression. Therefore, to avoid confusion for readers, we have revised that paragraph and highlighted accordingly. (Line 169-170) 

Q7: how did you check model fitness?

Author’s response: See our responses in Q5-Q6)

Q8: Is 19242 the sample size of both breast cancer and cervical cancer, if so how did you merge since the two outcomes were measured differently and it could have different missed value, different response and such like…please put the sample size of each outcome variables

Author’s response: Both outcome variables had the same samples collected. We have addressed this in Table 2. .

Q9: Operational definition of some independent variables could be necessary like wealth index, media exposure…..

Authors’ response: We have revised and highlighted in the revised manuscript accordingly.

Q10: Is method of screening and frequency of screening not your concern?

Author’s response: Yes, we do such as due to recall bias and social desirability. Therefore, we address these in our limitation. 

Q11: is media access and media exposure the same in your study?

Authors’ response: We clarified this in Q4 abstract section and we have revised it to be consistently over the manuscript where relevant

Q12: What about the women already positive for breast cancer, and cervical caner and women age above 49 years?

Authors’ response: The DHS just collected data about ever breast cancer and cervical cancer screening in women aged 15-49 years old. As far as my knowledge, DHS did not collect data about breast and cervical cancer status.

Results

Q1: The mean age were calculated but reported with standard deviation why?

Authors’ response: In descriptive statistics, continuous variables such women's age was calculated as mean and SD. We have revised this in the text. 

Q2: In your method section, you wrote that 19242 women were included for the final analysis, while in the result section, 19251 women were included. Why does this discrepancy occur (19242 VS 19251)?

Authors’ response: 19242 women were observed numbers, and 19251 women were weighted frequency used in DHS. For consistency, we have revised this to reflect our analysis and highlighted in Statistical analysis (Line 118-119).

Q3: How smoking was measured (smoking in last ten years, is smoking last 20 years or smoking on data collection) which one?

Authors’ response: In DHS, the smoking variable was asked as “Current tobacco use status at the time of survey”.

Q4: In alcohol drinking you classify never drink, ever drink and currently drink if so currently drink is included in ever drink individuals, how did you manage it?

Authors’ response: Current drink included only at least one drink in the last month. 

Q5: You calculated Odds ratio but you say risk factor, how?

Authors’ response: We revised it accordingly, as highlighted in the text.

Q6: There is new variable in your result section, how?

Authors’ response: We have already re-checked, and we do not include any new variable.

Discussion

Q1: You did not measure awareness about breast self-examination

Author’s Response: Awareness about breast self-examination is not included in the CDHS 2022 questionnaire.

Q2: Please write the implication of your finding?

Authors’ response: We revised it accordingly, as highlighted in the text.

Conclusion

Q1: write result specific conclusion and recommendation

Authors’ response: We revised and highlighted in the revised manuscript accordingly.

Write the authors contribution of the study

Author’s Response: We will do so followed the PLOS ONE guideline including funding, authors' contributions, data availability, and competing interests after the main manuscript. 

Reviewer #2: 

The manuscript is very interesting and contains significant information on breast and cervical cancer screening across the reproductive age group and it is worthy for publication.

However, there are few minor suggestions which I would like to suggest.

1. In tittle, I would suggest that replace the "Data analysis" as "A secondary analysis/ A statistical analysis".

Authors’ response: We thank the reviewer for this feedback. We revised it accordingly.

2. In Table-2, It is well understood with only Yes (%) out of total sample of breast and cervical cancer screening. Here you can add the chi-square value.

Authors’ response: We thank the reviewer for this feedback. We revised it accordingly.

---

## [Decision Letter · Decision Letter 1]

2 Jul 2024

PONE-D-23-39984R1Breast and cervical cancer screening among women at reproductive age in Cambodia: A Secondary Analysis of Cambodia Demographic and Health Survey 2022PLOS ONE

Dear Dr. Um,

Thank you for submitting your manuscript to PLOS ONE. After careful consideration, we feel that it has merit but does not fully meet PLOS ONE’s publication criteria as it currently stands. Therefore, we invite you to submit a revised version of the manuscript that addresses the points raised during the review process.

Additional Editor Comments:

Thank you for submitting the revised manuscript. Although you have made efforts to address most of the reviewers' comments, they are not yet fully satisfied with the revisions. The reviewers have asked for additional explanations and further revisions in certain sections of the manuscript. They have also suggested that the manuscript undergo a thorough English language edit or professional proofreading to improve clarity and readability.

Please make sure that these concerns are fully addressed and that the manuscript is thoroughly proofread before your next submission. We look forward to receiving your revised manuscript.

Thank you.

We look forward to receiving your revised manuscript.

Kind regards,

Dr Pijush Kanti Khan, Ph.D.

Academic Editor

PLOS ONE

Reviewers' comments:

Reviewer's Responses to Questions

**Comments to the Author**

1. If the authors have adequately addressed your comments raised in a previous round of review and you feel that this manuscript is now acceptable for publication, you may indicate that here to bypass the “Comments to the Author” section, enter your conflict of interest statement in the “Confidential to Editor” section, and submit your "Accept" recommendation.

Reviewer #1: (No Response)

Reviewer #3: All comments have been addressed

2. Is the manuscript technically sound, and do the data support the conclusions?

Reviewer #1: Partly

Reviewer #3: Yes

3. Has the statistical analysis been performed appropriately and rigorously? 

Reviewer #1: Yes

Reviewer #3: Yes

4. Have the authors made all data underlying the findings in their manuscript fully available?

Reviewer #1: No

Reviewer #3: Yes

5. Is the manuscript presented in an intelligible fashion and written in standard English?

Reviewer #1: No

Reviewer #3: Yes

6. Review Comments to the Author

Reviewer #1: Dear esteemed editor,

Thank you for providing me with the opportunity to review this manuscript. I would like to extend my appreciation to the authors for submitting a revised version of their manuscript. However, I would like to request that the authors provide a more detailed and elaborative response to the comments and suggestions provided in the previous review. Totally I am not satisfied with the revision. While I appreciate the effort made by the authors in their revision, I found that the point-by-point response in the submission documents was insufficient in addressing all of the concerns raised in the previous review. Therefore, I would like to request that the authors provide a more detailed and comprehensive response, addressing each of the comments and suggestions in a clear and concise manner.

Furthermore, I would like to request that the authors pay careful attention to the language used in their revision. It is important that the manuscript be written in a clear and concise manner, with proper grammar and syntax. This will ensure that the manuscript is easy to read and understand for all readers.

Thank you for your attention to these matters. The current comments are raised section by section below.

Abstract

1. On method section the likelihood measures should be written.

2. On result section it says the pooled prevalence from which did you pooled as we know DHS is secondary data, please clarify how did you pooled, and is only prevalence is pooled or factor variables also pooled?

3. On conclusion you conduct research on Adherence, while you conclude IFAS why?

Introduction

IFAS is one of strategy for morbidity and maternal mortality reduction due anemia. There is many articles and many systematic reviews conducted in previously so briefly show what is methodological gap, empirical gap, evidence gaps and knowledge gaps you could be filled in your research.

Method

1. Why 9 countries since East Africa isnot only 9

2. Ethiopia had demographic survey of 2019, why 2016?

3. Variable measurement needs reference

4. Result need more elaboration, in one area it says determinate, while in title it says factor, is it similar?

5. Discusion add the implication

6. In conclusion IFAS compliance was limited to one-third of pregnant mothers, what does it mean?

7. Conclusion is putting the result, please revise the conclusion

8. In conclusion the reader expected the recommendation but there is no single sentence which recommend based on your findings, why?

9. Reference 1, 2,10,18, is not full

Reviewer #3: Thank you for allowing me to review this interesting work. The authors have effectively responded to the comments. I recommend publishing this article after thorough proofreading.

7. PLOS authors have the option to publish the peer review history of their article (what does this mean?). If published, this will include your full peer review and any attached files.

Reviewer #1: No

Reviewer #3: No

---

## [Author Response · Author response to Decision Letter 1]

16 Oct 2024

Dear Dr. Emily Chenette, 

We would like to express our sincere gratitude to the editor-in-chief, academic editor, and reviewers for the significant time and effort put into further reviewing our manuscript. Upon reviewing the comments provided in revision R1 of reviewer #1, we have determined that the majority of these comments do not pertain to the content of our manuscript. Given this, we believe that the feedback may have been intended for a different submission. We respectfully request that the review comments be reconsidered in the context of our manuscript. We are committed to addressing any relevant concerns and are happy to provide further clarification if needed.

Kind regards,

Samnang Um, PhD

National Institute of Public Health

Phnom Penh, Cambodia

---

## [Editor Report · Decision Letter 2]

3 Jan 2025

Breast and cervical cancer screening among women at reproductive age in Cambodia: A Secondary Analysis of Cambodia Demographic and Health Survey 2022

PONE-D-23-39984R2

Dear Dr. Um,

We’re pleased to inform you that your manuscript has been judged scientifically suitable for publication and will be formally accepted for publication once it meets all outstanding technical requirements.

Kind regards,

Kazunori Nagasaka

Academic Editor

PLOS ONE

Additional Editor Comments (optional):

Dear Authors,

Thank you very much for submitting your manuscript to PLOS ONE.

After careful evaluation and consideration of the expert reviewers' conclusions, I am pleased to inform you that your manuscript has been accepted for publication.

Thank you again for your efforts and contribution to the field.

Sincerely,

Kazunori Nagasaka
---

## [Editor Report · Acceptance letter]

22 Jan 2025

PONE-D-23-39984R2 

PLOS ONE

Dear Dr. Um, 

I'm pleased to inform you that your manuscript has been deemed suitable for publication in PLOS ONE. Congratulations! Your manuscript is now being handed over to our production team.

Kind regards, 

on behalf of

Professor Kazunori Nagasaka 

Academic Editor

PLOS ONE